# Continuous Mechanical Extraction of Fibres from Linseed Flax Straw for Subsequent Geotextile Applications

**Saif Ullah Khan** [1,2,3], **Laurent Labonne** [1], **Pierre Ouagne** [3] and **Philippe Evon** [1,*]

1 Laboratoire de Chimie Agro-industrielle, Université de Toulouse, INRAE, ENSIACET, 31030 Toulouse, France; saifullah.khan@toulouse-inp.fr (S.U.K.); laurent.labonne@toulouse-inp.fr (L.L.)

2 Department of Textile Engineering, Balochistan University of Information Technology, Engineering and Management Sciences, Quetta 87300, Pakistan

3 Laboratoire Génie de Production, Université de Toulouse, ENIT, 65016 Tarbes, France; pierre.ouagne@enit.fr

* Correspondence: philippe.evon@toulouse-inp.fr

**Abstract:** Linseed flax is a multipurpose crop. It is cultivated for its seeds and particularly for its oil. The main contributors for this crop are Canada, France and Belgium. In general, straws of linseed flax are buried in the fields or burnt. However, these solutions are not good practices for the environment and from an economical point of view. In this study, straws of linseed flax (six batches in total) with different dew retting durations and harvesting techniques were studied to possibly use them for producing innovative geotextiles. Two different fibre extraction processes were investigated. A first process (A) involved horizontal breaker rollers and then a breaking card. A second one (B) consisted in using vertical breaker rollers, and an "all fibre" extraction device (fibre opener) followed by sieving. The chemical composition of fibres in parietal constituents appeared to be globally equivalent to the one of textile flax with a pectic content decreasing as a function of the dew retting duration. This contributed to an increase in the cellulose content. The fibre content was situated in a range from 29% to 33%, which corresponds to a good yield for linseed flax fibre. The level of purity can reach values of up to 90% for method A (without extra-sieving) and 96% for method B (with extra-sieving), and the length of the fibres (larger for method A than for method B) and their tensile properties make them suitable for structural geotextile yarn manufacturing.

**Keywords:** linseed flax; straw; fibre mechanical extraction; shives; mean fibre length; mean fibre diameter; tensile properties; geotextiles

## 1. Introduction

Between 2016 and 2018, an average of 97,700 ha/year of textile flax was cultivated in France with an average straw yield of 6 t/ha [1]. So, about 600 ktons of straw are globally available on this market. This economic activity represents about 12,000 direct jobs in in the growing and scutching sectors [2]. The long line fibres (about 25% of the straw mass) are used for fine and delicate textiles and structural composite materials, which are in high demand. In 2015, China was importing about 140 ktons of these fibres, representing approximately the total amount of the French production [3]. Even if the long textile flax are also produced in Belgium and the Netherlands, the demand is continuously increasing from China and Europe, but the land available for the production of textile flax cannot globally be extended in a great extent. The demand for tow fibres (scutching and hackling tows) is also very high for the manufacturing of coarser yarns for home textiles or technical applications such as in the composite industry for medium load bearing applications or for injected parts [4].

To answer the high flax fibre demand (both long line and tows), other sources of fibres may be promoted. Linseed, or oilseed flax may be one of them, as at the present time the valorization of this straw and its associated fibres only concerns low added value applications such as insulations or paper, as the traditional linseed flax straw is often

short in length (40–60 cm) with a high ramification, leading to a low straw yield (0.4 t/ha) available for fibre production [5]. Rennebaum et al. [6] showed that much higher straw yields can be obtained if no chemical shortener is used. A straw yield of 5.1 t/ha was obtained in their study, and this indicates that the linseed flax straw in their study consisted of between 25% and 30% fibre (in weight), and so could be a large source of fibre. The tensile properties of linseed flax fibres were also investigated by Pillin et al. [7]. Their work showed that the tensile properties of elementary fibres extracted manually are equivalent to the ones of textile flax extracted using the same process. During the continuous mechanical extraction of linseed flax fibres, Ouagne et al. [8] showed that the process parameter should be chosen with care as this may degrade the mechanical potential of the fibres if the extraction process is too aggressive.

An "all fibre" extraction set-up needs to be used as the linseed flax stems are randomly aligned on the ground and collected as such in round bales. In those conditions, it is not possible to use the same device as the one used for textile flax (scutching and hackling). Other devices exist to extract fibres from linseed flax or hemp stems. These ones are often based on hammer mills, which is a very efficient process permitting to break the woody part of the stem and to separate globally the shives from the fibres [9]. The process is very efficient but also very aggressive, and it was demonstrated that the fibres extracted using this process show relatively poor mechanical properties [10]. It is therefore important to minimize the impact of the fibre extraction on the mechanical potential of the fibres, especially when a geotextile valorization is targeted [11,12]. With hammer mill fibre extraction, the fibres are damaged and numerous defects such as kink-bands that are at the origin of strength and modulus decreases may be observed [13,14]. Two other processes which are expected to minimise their impact on the fibre mechanical properties will be tested in this study.

During fibre extraction, most of the shives in the straw are separated [8]. As for shives from textile flax, they could be used as animal litters because of their high water absorbency [15]. Recent works have nevertheless shown that shives from linseed flax could be also used to replace wood particles in the manufacture of boards [15,16]. Extrusion-refining pre-treatment of shives increases their mean aspect ratio and, consequently, the mechanical strength of the boards, and proteins from linseed cake can even be added as naturel binder to further increase this strength [16]. On the basis of their usage properties, such materials could find various applications, e.g., intermediate containers, furniture, domestic flooring, shelving, general construction, etc. In addition, as they do not emit formaldehyde, they are much more environmentally and human-health friendly than the wood-based boards currently found in the market, e.g., medium-density fibreboard (MDF), chipboard, oriented strand board (OSB) or plywood [16].

The aim of the present work is to study and to compare two different fibre extraction methods used to obtain technical fibres from various batches of oleaginous flax straw having different characteristics (e.g., cultivation locations, cutting height, and dew retting duration), for subsequent geotextile applications. Fibre content, fibre purity, and the mechanical properties of mechanically extracted fibres are determined, and they are discussed in relation to field parameters such as dew retting performed to ease the fibre extraction and to determine if the geotextile targeted application may seriously be considered to increase the income of the farmers besides the ones of seeds (for human or industrial oils) and shives (for animal litters and boards) [15,16].

## 2. Materials and Methods

### 2.1. Plant Material

Straws of oleaginous flax (*Linum usitatissimum* L.) used in this study were supplied by Ovalie Innovation (Auch, France). They were from the Everest variety, and the plants were cultivated in the south-west part of France. All the straw samples tested were collected using a combine harvester, and their harvesting took place after the seed one. Six different batches were used, with different dew retting durations (i.e., from 0 to 6 weeks) and with

different harvesting methods (i.e., high cut or low cut of the straw, and disconnection or not of the shredder of the combine harvester during the straw harvest). The details of those six batches are given in the Table 1. On the one hand, the cutting height of the straw directly influenced the quantity of straw harvested in the field. For a high cut (i.e., 30 cm above the ground), the length of the harvested stems was on average 35 cm. For a low cut (i.e., only 5 cm above the ground), it was around 60 cm in length. On the other hand, the objective of the disconnection of the shredder of the combine harvester during the straw harvesting was to better preserve the fibre integrity.

**Table 1.** Harvesting details of all the batches used in this study.

| Batch Number | Cutting Type | Disconnection of the Shredder | Dew Retting Duration before Harvesting |
|:---:|:---:|:---:|:---:|
| 1 | High cut | Yes | 6 weeks |
| 2 | High cut | Yes | No dew retting |
| 3 | Low cut | No | No dew retting |
| 4 | Low cut | Yes | No dew retting |
| 5 | High cut | No | 2 weeks |
| 6 | Low cut | No | No dew retting |

Those six batches were cultivated during the summer 2018 in two different locations. However, both were relatively close to each other (14 km). Firstly, batches number 1 to 4 were collected in the city of Pavie (43°62′ N, 0°57′ E). For this first location, after the seeds were harvested on July 11, the straw batches number 2 to 4 were harvested on July 12 and the straw batch number 1 was harvested on August 22, i.e., after six weeks of dew retting. Secondly, batches number 5 and 6 were collected in the city of Seissan (43°49′ N, 0°55′ E). For this second location, the seed harvesting took place on August 12, and straw batches number 5 and 6 were made on August 28 (dew retting during two weeks) and August 16 (no dew retting), respectively.

### 2.2. Manual Extraction of Fibres

For all the batches of oleaginous flax straw treated in this study, around fifty stems were chosen randomly, and those were used to separate manually fibres from shives. At the end of the manual extraction, the mass content of fibres inside each batch was determined. In parallel, the fibres extracted were used for determining their chemical composition.

### 2.3. Chemical Composition of Fibres

The moisture content of the fibrous materials was determined according to the ISO 665:2000 standard [17].

The chemical composition of fibres was determined from those which were extracted manually. Once extracted, the fibres were cut with a pair of scissors to a length of about 4–5 mm. Then, they were grinded finely using a Foss Tecator Cyclotec 1093 mill (Foss, Hillerød, Denmark) fitted with a 1 mm sieve. The obtained powder was the test sample used for the chemical characterizations.

Firstly, the three main parietal constituents, i.e., cellulose, hemicelluloses, and lignins, were evaluated using the ADF-NDF (ADF for Acid Detergent Fibre, and NDF for Neutral Detergent Fibre) method of Van Soest and Wine [18,19]. A Foss Tecator FT122 Fibertec hot extractor (Foss, Hillerød, Denmark) was used for the hot extractions, and a Foss Tecator FT121 Fibertec cold extractor (Foss, Hillerød, Denmark) was used for the cold ones.

Secondly, pectins were also quantified inside the manually extracted fibres. To begin, a non-normalized two-step extraction procedure was conducted on the test sample for analyses using each time a Foss Tecator FT122 Fibertec hot extractor:

(i)  An alcohol-insoluble solid fraction was first obtained after 3 min boiling in ethanol (95°), filtration, rinsing with ethanol (80°), and then drying.

(ii)  To continue, the total pectins were extracted, and their acid hydrolysis into galacturonic acid was realized; for these two simultaneous actions, 30 min boiling in 100 mL HCl 0.05 M was required, and the extraction was conducted twice.

Once the extract containing galacturonic acid recovered, it was then analysed using the Blumenkrantz and Asboe-Hansen colorimetric method for quantifying galacturonic acid [20]. Pectins are in fact polymers of galacturonic acid, i.e., polygalacturonic acids.

For each replication, the test sample mass was around 1 g for the determination of cellulose, hemicelluloses and lignins, and it was around 500 mg for measuring pectins. All determinations were conducted in duplicate. Results were expressed as mean values ± standard deviations. They were mentioned as a percentage of the dry matter mass of the fibres.

### 2.4. Mechanical Extraction of Fibres

Two different methodologies were used in this study to extract mechanically fibres from the oleaginous flax straw batches, i.e., the method A, and the method B. The two next paragraphs describe them in more details.

### 2.4.1. Method A

The method A consisted in the succession of (i) horizontal breaker rollers (Hemp-Act, Lacapelle-Marival, France), (ii) a thresher (Hemp-Act, Lacapelle-Marival, France), and then (iii) a breaking card (Hemp-Act, Lacapelle-Marival, France). In the horizontal rollers, the straw samples were broken, and shives were partly removed by gravity at the bottom of the device. The latter was in fact the assembly of five horizontal rollers in a staggered arrangement. At the output of these horizontal rollers, the fibrous material was then fed to the thresher, where additional shives were removed by intense threshing. Lastly, a breaking card from Russian origin inspired from the classical Mackie breaking cards was used for an additional removing of shives plus the alignment of fibres. At the outlet of the breaking card, a web was formed, and it was rolled up continuously.

### 2.4.2. Method B

The method B consisted in the succession of (i) vertical breaker rollers (Taproot Fibre Lab, Port Williams, NS, Canada), (ii) an "all fibre" extraction device (Laroche, Cours, France), and then (iii) a sieving machine (Ritec, Signes, France). To begin, the straw samples were processed in the vertical breaker rollers of a lab-scale device (processing machine) after stems (around 1 kg) were packed inside small bags having an appropriate volume. Here, the main purpose of this step was to break the stems, and to remove part of the shives inside the oleaginous flax straw before mechanically extracting the fibres. There were five vertical breaker rollers, and an efficient breaking of the stems was obtained thanks to their specific groove structure, to their staggered position, and also to an optimized spacing between them. After the breaker rollers partially broke the straw samples, the latter were kept in a climatic chamber at 25 °C and 90% relative humidity (RH) during seven to ten days to favour the adsorption of water by fibres until equilibration (i.e., up to a constant weight). Because this conditioning step allowed an increase in the moisture content of the fibrous material up to 14–19% in weight, it was helpful for the next fibre extraction step to better preserve the fibre length during their mechanical extraction. Indeed, with an increased moisture content, Ouagne et al. [8] evidenced that the rigidity of fibres is reduced, thus leading to less breakages when extracting them mechanically.

Then, the fibrous material was treated using a Laroche Cadette 1000 "all fibre" extraction device (Laroche, Cours, France). This industrial equipment was used to extract fibres in a mechanical way, and it is presented in Figure 1. With a 1 m width, this machine allowed the treatment of a 175 kg/h inlet flow rate of the fibrous material collected at the outlet of the vertical breaking rollers, corresponding to a 3.5 m/min speed for the feed belt.

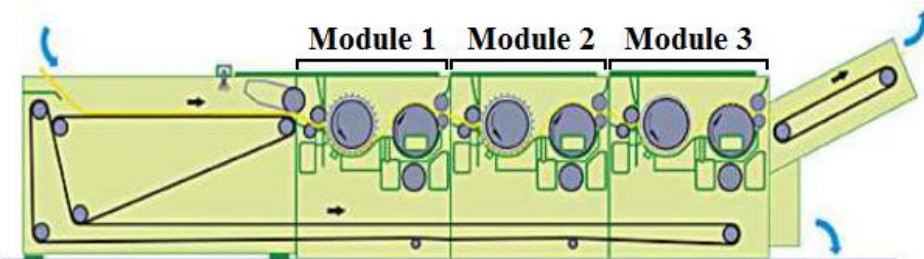

**Figure 1.** Laroche Cadette 1000 "all fibre" extraction device used in this study (from Laroche Cadette company website, https://laroche.fr, accessed on 21 June 2021).

This tearing machine has a double ability, i.e., the opening and the cleaning of natural fibres, as well as the formation of laps. It is composed of three successive modules. A pair of rollers, one smooth and the other grooved (made of rubber), is placed at the inlet of each module, thus ensuring a regular feeding of the raw material. In each module, a first cylinder equipped with nails (i.e., the extracting roller or fibre extraction roller) was the place where fibres were extracted. Its rotation speed can vary from 750 rpm to 1800 rpm. In addition, at the level of each of the three modules, shives can be (at least partially) eliminated by gravity, thanks to a trap door situated under the extracting roller. In this study, the opening of the trap door was set at its maximum to favour the most possible the removal of shives.

A perforated cylinder is also positioned at the end of each module. It is equipped with a ventilation system having a motor with a 2865 rpm maximum rotation speed. The perforated cylinder has three functions: (i) removing dust from the fibrous material by aspiration, (ii) forming a fibrous lap, and (iii) passing it on to the next module or the machine outlet.

The operating parameters used in the present study for the "all fibre" extraction device are presented in Table 2. They were the same as those used in [8], as the extraction of fibres from oleaginous flax straw was considered in that previous study as satisfactory for the two batches tested. Finally, the speed of the output belt was 1.8 m/min.

**Table 2.** Operating parameters used for the "all fibre" extraction device.

| Operating Parameter | Module 1 | Module 2 | Module 3 |
|---|---|---|---|
| Transmission speed of the lap (m/min) | 2.2 (from module 1 to module 2) | | 1.5 (from module 2 to module 3) |
| Rotation speed of the extracting roller (rpm) | 725 | 725 | 725 |
| Rotation speed of the motor for the aspiration at the level of the perforated cylinder (rpm) | 1500 | 2000 | 2000 |

Lastly, after the mechanical extraction of fibres using the "all fibre" extraction device, the fibrous lap obtained was sieved using a Ritec 600 (Ritec, Signes, France) vibrating sieving machine fitted with a 12 mm sieve. Here, the objective was to eliminate as much as possible the dust and the residual shives still trapped inside the lap, to obtain oleaginous flax fibres with a much higher purity.

*2.5. Purity in Fibres of the Different Fibrous Materials*

As the laps collected after the mechanical extraction of fibres did not consist only in bast fibres but also in some remaining shives and dust, the content in those impurities inside the different fibrous materials (including the starting oleaginous flax straw batches) was evaluated. For each fibrous material, a 50 g test sample mass was collected, and shives and dust were removed thanks to a mechanical sieving operation conducted during 5 min.

Then, the residual shives that remained trapped were collected manually. The content of impurities inside the fibrous material, and thus the real bast fibre content could then be calculated, and they were expressed as a percentage in weight.

### 2.6. Morphological Analysis of the Mechanically Extracted Fibre Bundles

The morphological analysis of the vegetal fractions obtained at the outlet of the fibre extraction devices used in this study were carried out on all the treated batches. A specific mass (around 10 g) of the extracted fibres was randomly picked from different places inside the laps, and around two hundred fibre bundles were measured in length.

Each bundle was attached to one end and then linearly extended in order to know its actual length, which was measured between its two extremities using a double decimetre with a 0.25 mm accuracy. The average fibre length was then calculated, and also the corresponding standard deviation.

### 2.7. Tensile Testing on Elementary Fibres

Diameter, tensile strength and elastic modulus of elementary fibres were measured using a Diastron Ltd. Lex 820 (Diastron Ltd., Andover, UK) automated high-precision extensometer. For this purpose, elementary fibres were firstly separated, and they were then glued on a 12 mm gauge length. To begin, the samples were placed on a peripheral equipment for the diameter measurement. This portion of the Diastron testing machine measured the diameter of each elementary fibre along ten different places throughout the whole fibre length. The diameter was measured with an accuracy of 0.01 μm.

Then, the same fibre sample was positioned in the high-precision extensometer of the testing machine for measuring tensile strength and elastic modulus. Here, a stepping motor was used for traction up to the breakage of the individual fibre, and a 20 N capacity load cell was used for measuring the force applied to the sample analysed. The extensometer was used for failing at low strain values. The accuracy for the obtained displacement was 1 μm, and the tensile test was conducted at a 1 mm/min speed. A 20 ms periodicity was chosen for recording the measuring points, i.e., the force and the displacement. The calculation of the tensile strength and elastic modulus thus became possible without using any supplementary strain measurement device. For each batch, forty samples of individual fibres were tested, and the mean value and standard deviation were then calculated for both tensile strength and elastic modulus.

## 3. Results

### 3.1. Contents in Fibres and Shives inside the Straw Batches

Table 3 mentions the contents of fibres and shives inside the six straw batches tested in this study. A manual extraction of the fibres inside the straw samples was conducted to obtain these results. For batches number 1 to 5, the content of fibres inside straw varied from 29% to 33%, and these results were in perfect accordance with those of a previous study where the fibre content inside linseed flax straw was found to be around 30% as well [21]. In another study [6], the fibre content inside technical stems was a little lower (from 23% to 29%), depending on different factors such as the variety of oleaginous flax.

**Table 3.** Contents in fibres and shives in the different straw batches tested (determination through manual extraction) (% in weight).

| Batch Number | Fibre Content | Shives Content |
|:---:|:---:|:---:|
| 1 | 30.1 | 69.9 |
| 2 | 28.9 | 71.1 |
| 3 | 33.3 | 66.7 |
| 4 | 32.8 | 67.2 |
| 5 | 32.2 | 67.8 |
| 6 | 45.9 | 54.1 |

When comparing batches number 3 and 4 with batches number 1 and 2, all coming from the same location, a low cut used during the stem harvesting had a slightly positive effect in the fibre content of the starting straw, i.e., 33% instead of 30–31%. Because the fibre proportion along the stem is known to be a little more important at its bottom [22], choosing a low cut for the combine harvester thus favours the collection of a higher proportion of fibres. However, such a setting for the harvesting machine may cause machine breakage if stones are present in the field that are caught at the same time. The harvesting must therefore be done with great care in this case.

With a fibre content surprisingly much higher (i.e., 46%), the straw batch number 6 was the exception in this study. Here, the reason is presumably due to cultivation considerations. Indeed, for this specific batch, no plant-growth regulator was used on the plot during the oleaginous flax development, and stems partly suffered lodging before the seed harvesting. The subsequent straw harvesting using the combine harvester was thus much more difficult in that case, and it is reasonable to assume that some part of the shives were in fact directly lost on the field during the harvesting of the straw. Indeed, for this specific batch, it was observed before the manual extraction of fibres that stems were already partially broken, and shives were absent in some places along them.

In fact, with a 46% fibre content inside straw, batch number 6 was more in accordance with results on textile flax presented in another study [23], where the crude fibre content varied from 36% to 50%, depending on many factors such as the soil type, and the nature of the fertilizer applied. Nevertheless, one should be aware that this batch is not representative of what could be produced on average at an industrial scale.

### 3.2. Chemical Composition of Bast Fibres

With cellulose, hemicelluloses and lignins, pectins are one of the main constituents of the plant cell walls. They are also the predominant compound within the middle lamella. In fact, pectins have the function to hold the cells of plant tissue together. In linseed flax stems, pectins are more specifically responsible for "cementing" the bast fibres together. Thus, by reducing the pectin content, this definitely helps to separate the fibres from each other, and both their extraction and opening thus become facilitated. For fibrous plants such as textile flax, linseed flax and hemp, this is the retting operation that allows the removing of pectins before conducting the extraction of fibres, and the evolution of their content over time is a clue to estimate its efficiency.

Table 4 shows the evolution of the pectin content inside manually extracted bast fibres as a function of the dew retting duration. Dew retting is a natural retting process conducted directly on the ground before the straw harvesting. It involves in fact two different actions, both resulting in the progressive dissolution of the pectic binder over time, i.e., a chemical action (thanks to the morning dew) and a biological one (thanks to the microorganisms, fungi and bacteria in soil).

**Table 4.** Chemical composition of manually extracted fibres as a function of the dew retting duration (from 0 to 6 weeks) (% of dry matter).

| Dew Retting Duration | Hemicellu-Loses | Lignins | Cellulose | Pectins |
|:---:|:---:|:---:|:---:|:---:|
| No dew retting (i.e., batch number 2) | 12.9 ± 0.5 | 5.3 ± 0.4 | 62.1 ± 0.4 | 4.4 ± 0.2 |
| 2 weeks | 6.6 ± 0.1 | 7.2 ± 0.4 | 65.5 ± 0.4 | 4.4 ± 0.1 |
| 4 weeks | 8.5 ± 0.5 | 6.1 ± 0.3 | 66.9 ± 0.5 | 3.6 ± 0.2 |
| 6 weeks (i.e., batch number 1) | 10.3 ± 0.0 | 5.5 ± 0.1 | 67.0 ± 0.1 | 3.7 ± 0.2 |

Results are presented as mean values ± standard deviations.

For this part of the work, batches number 2 and 1 were analysed, corresponding to no dew retting and to a six weeks dew retting duration, respectively. To complete the analysis, two stem samples were also collected in the same plot during the retting period, i.e., after two weeks and four weeks of dew retting, respectively. Table 4 also mentions

the evolutions in the content for the three other parietal constituents (i.e., hemicelluloses, lignins, and cellulose).

From the results in this table, it is clear that the dew retting played an important role in the reduction of the pectin content, which was only 3.7% after six weeks of dew retting instead of 4.4% immediately after the seed harvesting (i.e., no dew retting). The pectin content remained unchanged after two weeks, and a four weeks dew retting duration was at least necessary to see a reduction in the pectin content inside the manually extracted bast fibres. In parallel, from zero to six weeks, their cellulose content increased progressively (i.e., from 62.1% to 67.0%), indicating that retting was effective but rather slow. The content of hemicelluloses was also lowered after retting, whatever its duration. In contrast, because lignins are biopolymers with a high hydrophobic character due to their polyphenol structure, their content inside bast fibres logically tended to increase during retting.

### 3.3. Purity of Extracted Fibres from Process A

The A methodology for the extraction of fibres from straw used a succession of three different operations, i.e., (i) horizontal breaker rollers, (ii) a thresher, and then (iii) a breaking card. No sampling was made from this methodology after the breaker rollers and the thresher, meaning that the purity of the extracted fibres was measured only inside the final fibrous material, collected at the outlet of the breaking card in the form of a rolled web. The purity of the obtained webs in fibres is mentioned in Table 5, and it varied from 66% to 90%.

**Table 5.** Comparison of the final purities in fibres (%) for the two processes tested (i.e., A and B).

| Batch Number | 1 | 2 | 3 | 4 | 5 | 6 |
|---|---|---|---|---|---|---|
| Process A | 89.9 | 66.9 | 79.2 | 66.2 | 84.7 | 87.8 |
| Process B | 89.3 | 91.9 | 88.5 | 81.9 | 93.8 | 96.1 |

The web originating from batch number 4 showed the minimum value for fibre purity (i.e., 66%). As a reminder, that straw batch was not dew retted, and the linseed flax straw was harvested using a low cut setting, with disconnection of the combine harvester shredder (Table 1). This resulted in longer stems with shives still well glued by the pectic cement to the bast fibres. Undoubtedly, this had reduced the chance to remove in a proper way shives from the extracted fibres. The web purity in fibres was in the same order of magnitude (i.e., 67%) for batch number 2. In the same way, this straw batch was not dew retted, and it was also harvested with the shredder of the combine harvester disconnected. On the contrary, the batch number 1 showed the maximal purity level (i.e., 90%) for the obtained web at the outlet of the breaking card. Because this batch was dew retted during six weeks, shives were less attached to the fibres. In addition, straw was harvested using a high cut in that case. Stems collected on the field were thus shorter, and shives were probably less retained by the fibre web during its formation in the inside of the breaking card. With that 90% purity, additional spinning and weaving operations should be perfectly possible to manufacture geotextiles. Inside the obtained webs, straw batches number 5 and 6 had quite comparable fibre purities to that of batch number 1, i.e., 85% and 88%, respectively. Additionally, this means that the corresponding extracted fibres would be also suitable for subsequent geotextile applications.

### 3.4. Purity of Extracted Fibres from Process B

As a reminder, the B methodology for the extraction of fibres from the straw involved three successive steps, i.e., (i) vertical breaker rollers for the stem breaking and the removing of part of shives, (ii) the "all fibre" extraction device for extracting fibres (plus the loss of more shives by gravity at the bottom of each of the three extracting rollers), and (iii) a sieving extra-step which consisted in eliminating as much as possible the remaining shives to increase more the purity in fibres of the final lap. Table 6 mentions the purity of the different fibrous materials in fibres after the vertical breaker rollers, after the "all fibre"

extraction device, and also after the sieving extra-step. The impurities in the fibrous lap after sieving also appear in Table 6. For the purpose of comparison between the two tested processes, the final purity in fibres for process B is also mentioned in Table 5.

**Table 6.** Purity of the fibrous materials in fibres during process B (i.e., after the vertical breaker rollers, for the fibre lap at the outlet of the "all fibre" extraction device, and after sieving), and amount of impurities in the fibrous lap after sieving (% in weight).

| Batch Number | After the Vertical Breaker Rollers | After the "All Fibre" Extraction Device | After Sieving | Impurities [1] in the Fibrous Lap after Sieving |
|---|---|---|---|---|
| 1 | 45.1 | 68.2 | 89.3 | 10.7 |
| 2 | 46.2 | 70.7 | 91.9 | 8.1 |
| 3 | 47.1 | 62.6 | 88.5 | 11.5 |
| 4 | 44.9 | 64.7 | 81.9 | 18.1 |
| 5 | 51.6 | 68.4 | 93.8 | 6.2 |
| 6 | 66.0 | 68.9 | 96.1 | 3.9 |

[1] Shives and dust amount.

After the fibre extraction step using the "all fibre" machine, a 63–71% purity in fibres was achieved instead of 29–46% in the starting materials. The minimal purities (i.e., 63% and 65% in weight, respectively) were observed in the case of the batches number 3 and 4, which were the non-retted batches and had a low cut at harvesting. Because stems were in that cases longer (i.e., around 60 cm) and still contained their pectic cement, the extraction of fibres was more difficult, and this resulted in a reduced fibre purity level. The separation of shives from fibres was disadvantaged due to the presence of more pectins (Table 4), and the shive particles really separated from the fibres remained also easier trapped inside the fibrous material made of longer stems. In parallel, when comparing batches number 3 and 4, the disconnection of the shredder of the combine harvester at the moment of the straw harvesting (case of batch number 4) had in fact no real importance for the purity in fibres at the "all fibre" device outlet. In fact, a low cut at harvesting should be privileged to increase the crop yield in the field, even if it is necessary in that case to pay attention to the increased risks of machine breakage.

The highest purity in fibres at the outlet of the "all fibre" extraction device (i.e., 71%) was obtained for the batch number 2. Because the latter was harvested using a high cut at the straw harvesting, stems collected were shorter, and woody parts were thus more easily removed (when harvesting straws just as when extracting fibres) as the stem entanglement was less significant. For the three other batches (i.e., batches number 1, 5 and 6), the fibre purities were quite similar and median, with contents of fibres between 68% and 70%. In particular, batches retted before the straw harvesting (i.e., batches number 1 and 5) did not reveal especially higher purity values after the "all fibre" extraction device. This means that even if the shives were more easily detached from the bast fibres once the straw had been retted, the "all fibre" extraction device was not able to remove them in totality from the inside of the obtained lap. In fact, part of them remained trapped inside the fibrous lap, and this is the reason why a sieving extra-step was applied to it to eliminate the remaining shives as much as possible.

After sieving, the purity of all batches was markedly increased, and it ranged from 82% to 96%, which was considered as a good material purity for future geotextile applications. In fact, such values confirmed the real efficiency of the sieving extra-step for reaching a better fibre purity of the final lap. In addition, the obtained purities were much better than those obtained from the A methodology using the breaking card as fibre extraction mode (i.e., maximum value of 90%). In the case of process B, the minimal fibre purity (i.e., 82%) was observed for batch number 4, and it was in fact exactly the same situation when using the A methodology for fibre extraction (Table 5). Although this purity would be enough for further processing in geotextile applications, the most likely reasons for such low purity may be the cutting type (i.e., a low one) on the straws on the field and the disconnection of

the combine harvester shredder during the straw harvesting. Those two harvesting settings both contributed to stems with higher length and thus in a more important entangling of fibres once extracted, and this probably unfavoured the removing of shives at sieving.

Batches number 5 and 6 showed the best values for purity (i.e., 94–96%). Here, it must be kept in mind that those two batches originated from a different location than the four others. Contrary to batches number 1 to 4, no plant-growth regulator was used during the linseed flax cultivation in that second location. It is thus reasonable to assume that both soil and climate conditions, and cultivation mode were the most likely reasons to explain such differences in fibre purity.

### 3.5. Morphological Analysis of the Extracted Fibre Bundles

Length of the extracted fibre bundles was measured after having randomly taking a handful of them from each batch. Then, around two hundred bundles were analysed for their lengths, and the results (i.e., mean values ± standard deviations) are given in Table 7. Here, the average length gives a good indication of the ease of subsequently obtaining a yarn, after passing over a drawing frame. For its part, Figure 2 represents the frequency distribution curve of the length of fibre bundles extracted from batch number 6 using the two extracting methodologies (i.e., A and B).

**Table 7.** Length of fibre bundles (cm) inside all the final laps produced.

| Batch Number | Process A | Process B |
| --- | --- | --- |
| 1 | 11.7 ± 6.2 | 9.4 ± 4.6 |
| 2 | 10.9 ± 5.9 | 6.4 ± 3.2 |
| 3 | 10.7 ± 5.6 | 7.7 ± 4.0 |
| 4 | 10.6 ± 5.7 | 8.3 ± 3.9 |
| 5 | 11.1 ± 5.3 | 8.1 ± 4.1 |
| 6 | 13.4 ± 7.8 | 7.9 ± 3.5 |

Results are presented as mean values ± standard deviations.

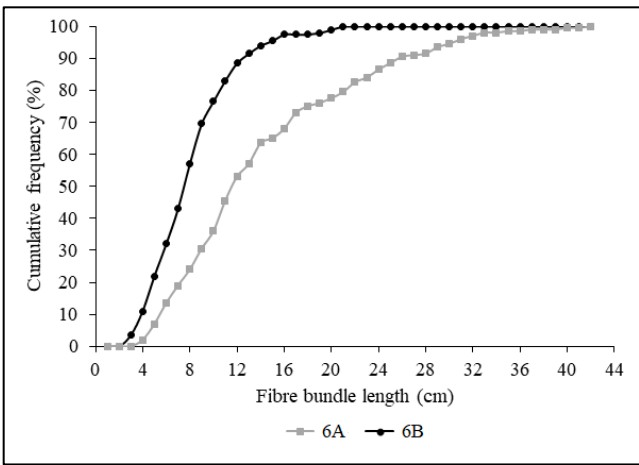

**Figure 2.** Frequency distribution curve of the length of fibre bundles extracted from batch number 6 using the A and B extracting methodologies.

As a first result, longer fibre bundles were obtained in the present study in comparison with other values from linseed flax fibres in the literature [8,24], regardless of the linseed flax straw batch and the extraction method used.

Additionally, Figure 2 perfectly illustrates the fact that process A is more accurate for better preservation of fibre length, and this was true for all six batches tested in this study, even though Figure 2 only shows the results obtained from batch number 6. Only 13% and up to 32% fibre bundles extracted from batch number 6 were observed less than 6 cm in length for processes A and B, respectively, which is interesting for further spinning stages.

Portions of 36% and 76%, respectively, of fibre bundles were less than 10 cm in length, and at least half of the fibre bundles (i.e., 50% and 62%, respectively) had a length comprised between 6 and 14 cm, which should be really interesting for subsequent spinning.

An increase in the standard deviations was nevertheless observed in the case of the fibre bundles extracted through process A (Table 7), thus indicating that a larger dispersion in length existed for the bundles extracted from the breaking card.

### 3.6. Mechanical Properties of Elementary Fibres inside the Extracted Bundles

The tensile properties of the individual fibres inside the extracted bundles are mentioned in Table 8. The individual fibres analysed were separated with care from the bundles using a method presented in [14]. Fibre bundles originating from the six straw batches used, and from both methodologies (i.e., A and B), were immersed in water at 30 °C for 72 h. Here, the objective was to facilitate the manual extraction of the elementary fibres from the bundles, and to extract them with the highest care.

**Table 8.** Tensile properties, minimal diameter, maximal diameter, and cross section area of individual fibres inside all the final laps produced.

| Characteristic | 1A | 1B | 2A | 2B | 3A | 3B | 4A | 4B | 5A | 5B | 6A | 6B |
|---|---|---|---|---|---|---|---|---|---|---|---|---|
| Tensile strength (MPa) | 456 ± 308 | 694 ± 374 | 832 ± 578 | 626 ± 419 | 778 ± 516 | 660 ± 383 | 751 ± 454 | 939 ± 525 | 701 ± 402 | 827 ± 490 | 874 ± 560 | 880 ± 1077 |
| Young's modulus (GPa) | 29.5 ± 14.1 | 39.4 ± 20.1 | 41.9 ± 24.0 | 35.3 ± 19.3 | 43.5 ± 22.4 | 36.7 ± 15.3 | 43.6 ± 23.8 | 46.2 ± 21.7 | 40.9 ± 21.7 | 50.6 ± 23.0 | 46.1 ± 24.8 | 63.7 ± 58.2 |
| Minimal diameter (μm) | 15.1 ± 5.7 | 16.2 ± 5.9 | 13.5 ± 5.7 | 15.4 ± 5.7 | 13.7 ± 5.5 | 16.1 ± 5.0 | 11.5 ± 3.8 | 12.1 ± 4.4 | 10.0 ± 3.1 | 10.5 ± 3.9 | 12.7 ± 5.8 | 13.0 ± 6.5 |
| Maximal diameter (μm) | 34.4 ± 16.2 | 33.4 ± 12.8 | 26.2 ± 10.9 | 31.8 ± 12.8 | 27.8 ± 11.1 | 31.2 ± 13.2 | 23.5 ± 9.1 | 29.2 ± 8.1 | 28.2 ± 10.4 | 25.2 ± 8.9 | 28.8 ± 12.0 | 30.4 ± 12.4 |
| Cross section area (μm$^2$) | 462 ± 338 | 466 ± 307 | 323 ± 253 | 431 ± 290 | 348 ± 271 | 442 ± 315 | 248 ± 201 | 306 ± 175 | 243 ± 159 | 241 ± 210 | 334 ± 302 | 364 ± 333 |

Results are presented as mean values ± standard deviations.

For all the treated samples, the section of the individual fibres had the form of an oval shape and not of a cylindrical one. Indeed, minimal and maximal diameters measured were always different. The first one ranged from 10.0 to 16.2 μm, whereas the second one ranged from 23.5 to 34.4 μm. The cross section area of the manually extracted individual fibres was thus between 241 and 466 μm$^2$, and this would correspond to diameters ranging from 17.5 to 24.4 μm if considering the cross section of all fibres as an ideal cylinder.

The mechanical results obtained for the elementary fibres inside the extracted bundles showed that a reduced cross-sectional area led to better tensile properties (Table 8). In addition, because the obtained diameter values were in the same order of magnitude as those mentioned in other studies for individual fibres from linseed [8] and textile [23] flax, it is reasonable to assume that, in the present study, fibres manually extracted from bundles using the hot water procedure were indeed elementary fibres.

Values for the average tensile strength ranged from 456 to 939 MPa. The minimal value (i.e., 456 MPa) was for the 1A sample, corresponding to the batch number 1 treated with the A methodology. In parallel, the maximal one (i.e., 939 MPa) was obtained from sample 4B (i.e., straw batch number 4 using the B methodology). Similarly, Young's modules ranged from 29.5 to 63.7 GPa (samples 1A and 6B, respectively).

## 4. Discussion

### 4.1. Contents in Fibres and Shives inside the Straw Batches, and Chemical Composition of Bast Fibres

The contents of fibres inside straw batches in this study (from 29% to 33% for batches number 1 to 5) are quite coherent in comparison with other data in the literature also dealing with oleaginous flax [21]. The batch number 6 was the only straw batch for which the fibre content was surprisingly much higher (i.e., 46%), and the reason was presumably due to cultivation considerations as mentioned above. Using oleaginous flax straw for its

richness in bast fibres (most often around 30% in weight with the exception of the 46% fibre content in the case of batch number 6, which is however not representative with what could be obtained at industrial scale) was thus quite possible, the straw possibly becoming in that way a supplementary added-value product of the linseed flax cultivation in addition to the seeds.

From the results in Table 4 presenting the chemical composition of bast fibres, it is obvious that dew retting contributed to a reduction in the pectin content. However, the duration for dew retting was probably not long enough in that case. The cultivation place in France (Gers department) is also known to be a quite sunny and hot region in summer, and with low rainfall. Surely, this can explain, at least in part, why the reduction in the pectin content was not so important. For future works, because dew retting is really of key importance, an increase in its duration will thus need to be tested in order to identify the optimal retting duration. Choosing agricultural plots at the bottom of the valleys, where the morning dew is stronger and more persistent over time, would also be better.

Jute and flax fibres [25] are subjected to the dew retting mostly. Pectin content is removed in dew retting due to bacterial activity. Dew retting depends on soil fungi colonization on stem/bast to degrade pectin and hemicelluloses by releasing polygalacturunase and xylanase.

In a review, ref. [26] reported the changes in flax fibre during dew retting process. Cell wall composition was directly related to the microbial colonization. Partial damage and fibre bundle decohesion was observed due to fungal hyphase and parenchyma on the epidermis and around the fibre during dew retting. Furthermore, a decrease on the primary cell wall of polysaccharides was noticed on the fourteenth day of retting due to higher enzymatic activities. This spreading of microbial colonization went towards the inner core of the stem after 6 weeks retting.

### 4.2. Comparison of the Purity of Extracted Fibres from Processes A and B

When comparing the two extracting methodologies, i.e., A and B, the batch number 4 resulted in both cases in the minimal fibre purity. This means that long stems for which no dew retting was applied before their harvesting were an unfavourable situation for reaching high fibre purity (Table 5). In contrast, batches number 5 and 6—for which no plant-growth regulator was used during the linseed flax cultivation—had high purity levels (up to 96% for batch number 6 treated using process B) once fibres were extracted, for both methodologies used.

In addition, except for batch number 1, laps produced using the B methodology (i.e., the "all fibre" extraction device plus the sieving extra-step) had all a more important fibre purity than the webs collected at the outlet of the breaking card for process A. This highlights with no doubt the real interest of the sieving extra-step conducted at the end of the B methodology. For future work, it could thus be interesting to apply the same sieving treatment to webs produced using the breaking card, on the condition of unwrapping them before sieving. In conclusion, producing geotextiles should be accessible with fibre purities of at least 85%, and such purity was in fact reached in eight out of twelve cases, including five out of six cases for laps originating from process B.

### 4.3. Morphology of the Extracted Fibre Bundles

The average fibre length for all batches (11–13 cm for process A, and 8–9 cm for process B) (Table 7) was increased as compared to previous values in the literature also obtained from linseed flax fibres, i.e., only 2 cm [24] and around 5 cm [8]. In the case of process B, this increase in fibre length in comparison with previous results [8] was probably partly due to the rewetting of the raw batches inside the humid climatic chamber before extracting fibres. With higher moisture levels (14–19%) for straw after conditioning instead of 9–10.5% before conditioning, the fibres thus became more flexible (i.e., less brittle), and their breakage during the severe extraction action in the "all fibre" extraction machine tended to be reduced. The same results were observed in Ouagne et al. [8] but to

a lesser extent. The overall fibre length for process B varied from 6.4 cm to 9.4 cm, which was considered as good enough for subsequent spinning process before obtaining the geotextiles. When looking more precisely to the length values for batches number 1 to 4, it has to also be mentioned here that a six weeks dew retting duration (case of batch number 1) tended to favour a better preservation of the bundle length, even if the differences were not statistically significant. On the contrary, for batches number 5 and 6 coming from the same plot, the lengths of bundles were the same, meaning that a two weeks dew retting was probably too short for generating longer fibre bundles once extracted.

Fibre bundles produced through the breaking card (i.e., process A) were much longer, and their average length varied from 10.6 cm to 13.4 cm. However, the associated standard deviations were even higher. In fact, the horizontal breaker rollers and/or the garniture of the breaking card (i.e., geometry, height and diameter of nails, and distance between them) used in process A were much more suitable to better preserve the fibre bundle length, which is surely a much less aggressive mechanical extraction technique in comparison with the "all fibre" machine used for process B. In addition, the interest of the dew retting step before the straw collection was less evident in the case of process A.

In fact, the highest average length of extracted fibre bundles coming from the A methodology was obtained for batch number 6, which was a non-retted batch. However, because a low cut setting was used at harvesting for that batch, the longer stems collected on the field favoured higher bundle length once extracted.

As a conclusion, with the longer fibres obtained from the breaking card (especially those from batch number 6), the subsequent spinning step will be easier from the A-based rolled web. Indeed, less twist should be required in that case, and there should have also more friction between the fibres, thus favouring better maintenance of fibre bundles in the form of slivers or yarns at weaving.

### 4.4. Mechanical Properties of Elementary Fibres inside the Extracted Bundles

Table 8 mentions the tensile properties of elementary fibres inside the produced laps. Here, the influences of (i) the batch type, and (ii) the mechanical extraction methodology on these properties were not easy to discuss. Nonetheless, when calculating a mean value of the average tensile strengths, it was 732 and 771 MPa for processes A and B, respectively. In parallel, this mean value was 41 and 45 GPa, respectively, for the average Young's modules. It is thus reasonable to consider that both mechanical extraction methodologies (i.e., the breaking card for process A, and the "all fibre" extraction device for process B) resulted in quite equivalent tensile properties for the individual fibres inside the extracted bundles. Such overall average values for tensile strengths and Young's modules will undoubtedly authorize the use of the extracted bundles in geotextile applications. Indeed, for comparison, a previous study dealing with the mechanical extraction of fibres from oleaginous flax straw revealed much lower tensile strengths for the individual fibres (i.e., only 323–377 MPa mean values) for the same application in geotextiles [8].

As a reminder, batch number 1 was dew retted for six weeks before the straw harvesting. However, the tensile properties of the individual fibres obtained from that batch were the minimal ones for process A (i.e., horizontal breaker rollers and breaking card route), and they were median for process B (i.e., vertical breaker rollers plus "all fibre" extraction device plus sieving). Whereas a long-term dew retting was expected to favour the tensile properties of the extracted fibres, the results of the present study from batch number 1 did not confirm surprisingly this assumption. On the contrary, the batch number 5 which was dew retted during only two weeks resulted in much higher tensile properties for both extraction processes tested. Nonetheless, the plots were not the same (i.e., differences in the soil composition and, to a lesser extent, in climatic conditions) for batches number 1 and 5. In addition, no plant-growth regulator was used in the case of batch number 5. It is thus difficult to conclude at this point on the differences observed in the tensile properties between these two batches.

However, in the case of textile flax, it should be borne in mind that dew retting can last up to three months depending on the climatic conditions and industrial requirements [27]. In addition, it is mainly grown in the Northern part of France, a region known to be much wetter and more prone to morning dew than the Gers department where the linseed flax straws in this study were produced. It is therefore reasonable to assume that the dew retting durations tested here (six weeks max) were simply not sufficient. For future work, longer durations should thus be tested. Due to the low rainfall observed in summer in south-western France, periods of at least three months should be considered. The plots should also be chosen carefully, and those at the bottom of valleys should be preferred because of their higher humidity.

In conclusion for this study, the elementary linseed flax fibres inside the extracted bundles thus revealed an average tensile strength of 750 MPa and an average Young's modulus of 43 GPa. This was in perfect accordance with recent results on the tensile properties of hemp fibres (660 and 38 GPa for tensile strength and Young's modulus, respectively) [14] and nettle ones (712–812 MPa and 36–53 GPa) [28], both intended for the production of load-bearing composite materials. As geotextiles require lower mechanical properties, the use of the linseed flax fibres resulting from this study should thus be possible for such an application.

The fibre extraction processes used in this study were selected to work with bast fibre type plants such as hemp, flax or nettle. As mentioned in the text, it is important that a minimum level of retting is performed so that the natural cements binding the fibres together and the ones binding the fibres to the rest of the plant are degraded. This favours the fibre division and the separation between the fibres and the woody core of the plant and the bark. As these machines are mainly dedicated to the fibre extraction of European bast fibres, one can imagine that they could also operate with other types of bast fibre plants such as kenaf or jute for example. Of course, modification of some of the settings would be necessary to accommodate the changes in plant diameter, or woody core breaking strength.

Both the machines were designed to operate in a dry state. If dried, resources such as banana trunks or sisal leaves could probably be used with the breaking rollers and the breaking card. However, some modifications in the corrugated rollers and of course in the process settings would probably be necessary. In any case, dedicated machines already exist for these kinds of resources [29,30], and the machines presented in this study should be dedicated to the use of retted bast fibre plants.

## 5. Conclusions

Linseed flax is a dual-purpose crop. Even if it is firstly cultivated for its seeds, its straw can be also useful, thus possibly contributing to an additional source of income for farmers due to its high bast fibre content. In this study, the straw batches tested had a significant amount of fibre, the latter varying from 29% to 46% of the straw dry mass. Depending on the dew retting duration of oleaginous flax straw on the field before its harvesting, the cellulose content inside bast fibres was between 62% when no dew retting was applied and 67% when a dew retting duration of six weeks was chosen. Simultaneously, their pectin content decreased from 4.4% to 3.7%. Dew retting appeared as an important parameter for the subsequent mechanical extraction of fibres and their opening. It favoured the extraction of fibres, contributing at the same time to a better preservation of the fibre length. Two different methodologies were also tested in this study for the fibre mechanical extraction. The first one consisted in the use of a breaking card, and the second one in the use of an "all fibre" extraction device. As the second methodology was completed by a sieving extra-step for better purifying the obtained fibre lap, it resulted in the production of purer fibres as compared to the first extraction method. However, the purity of the fibres extracted thanks to the breaking card was considered as good enough for geotextile applications, and this purity may be further increased by additional work through sieving and/or their treatment using new breaker rollers. For all the straw batches tested, the breaking card resulted in much longer fibre bundles, and this could be attributed to a too strong severity of the "all

fibre" extraction device when used. Looking at their length and their tensile properties, the extracted fibre bundles obtained from both methodologies would be of sufficient quality for further spinning into yarns. Thus, they appeared as promising candidates for the manufacture of innovative and renewable geotextiles.

For future work, much longer dew retting durations (at least three months or even more) should be applied as this is known to have a beneficial effect on the ease of the mechanical extraction of bast fibres. A low cutting height of the straw should also be favoured as it increases straw yields, especially if no shortener has been added during plant cultivation. The shredder of the combine harvester should also be disconnected as it allows a better preservation of the mechanical strength of the fibres at the moment of their harvesting. Lastly, with regard to the fibre extraction stage, a breaking card can produce longer fibre bundles, which can be an advantage for subsequent spinning operations. However, a sieving extra-step should be carried out in order to eliminate the residual shives present in the extracted fibres as much as possible.

**Author Contributions:** Conceptualization, P.E. and P.O.; methodology, P.E., P.O. and L.L.; validation, P.E. and P.O.; formal analysis, P.E. and P.O.; investigation, S.U.K. and L.L.; resources, P.E. and P.O.; writing—original draft preparation, S.U.K.; writing—review and editing, P.E. and P.O.; supervision, P.E. and P.O.; project administration, P.E. and P.O. All authors have read and agreed to the published version of the manuscript.

**Funding:** This research was funded by Région Occitanie, France, grant number MP0013559.

**Institutional Review Board Statement:** Not applicable.

**Informed Consent Statement:** Not applicable.

**Data Availability Statement:** Data is contained within the article.

**Acknowledgments:** The authors would like to express their sincere gratitude to Ovalie Innovation (Auch, France) for supplying the batches of oleaginous flax straw used for the purpose of this study.

**Conflicts of Interest:** The authors declare no conflict of interest.

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
