# Peer review of "Continuous Mechanical Extraction of Fibres from Linseed Flax Straw for Subsequent Geotextile Applications"

_coatings, doi:10.3390/coatings11070852_

Round 1

Reviewer 1 Report

The author study and to compare two different fibre extraction methods used to obtain technical fibres from various batches of oleaginous flax straw having different characteristics (e.g., cultivation locations, cutting height, and dew retting du-64 ration), for subsequent geotextile applications.

  1. The organization and presentation of the results should be improved. Figures about the linseed flax straw and products should be included.
  2. The composition of the fiber products from different methods should be analyzed.
  3. The advantages of the mechanical method used here could be should be compared with other papers.
  4. If this mechanical extraction is used for the treatment of the plant materials from other regions, what are important issues that should be considered?

Author Response

We thank Reviewer 1 for his helpful comments. Under are the answers to his questions.

The organization and presentation of the results should be improved. Figures about the linseed flax straw and products should be included.

Answer:

The structure of the paper was reorganized. The description of the results was positioned in the Results section (lines 256-474). And, the discussion one only focused on the scientific discussion, including the reference to the results of other studies (lines 475-757). Additionally, the Introduction section already mentioned that the linseed flax straw yield is of around 5 t/ha with a fibre content from 25% to 30% (in weight) (lines 51-53). However, additional comments on the use of shives as animal litter or for the manufacture of renewable boards was added in the revised version of the manuscript (lines 75-85).

The composition of the fiber products from different methods should be analyzed.

Answer:

The determination of the chemical composition of the bast fibres was only carried out on those that were manually extracted from the six batches of oleaginous flax straw used in this study. This composition was not determined for the mechanically extracted fibre bundles from processes A and B (i.e., 1A, 1B, ..., 6A, 6B samples), and such analyses are unfortunately no longer possible because we no longer have these fibres available in the laboratory. We apologise for this inconvenience.

However, as the extractions in this study were carried out through a simple mechanical procedure, there is no reason to have a difference in chemical composition between the manually extracted fibres (those that were chemically analysed) and those that were mechanically extracted.

The advantages of the mechanical method used here should be compared with other papers.

Answer:

As mentioned in the introduction part of the paper, an “all fibre” extraction set-up needs to be used as the linseed flax stems are randomly aligned on the ground and collected as such in round bales. In those conditions, it is not possible to use the same device as the one used for textile flax (scutching and hackling). Other devices exist to extract fibres from linseed flax or hemp stems. These ones are often based on hammer mills, which is a very efficient process permitting to break the woody part of the stem and to separate globally the shives from the fibres (Xu et al., 2012). The process is very efficient but also very aggressive, and it was demonstrated that the fibres extracted using this process show relatively poor mechanical properties (Placet et al., 2012). These ones are globally half of the ones obtained in this work.

With hammer mill fibre extraction, the fibres are damaged and numerous defects such as kink-bands that are at the origin of strength and modulus decreases may be observed (Baley, 2004; Grégoire et al., 2019) with a much greater extent than in the case of the breaking roller/breaking card.

A paragraph was added in the introduction Section to specify the reviewer’s point (lines 60-73).

Related references:

Xu, J.; Chen, Y.; Laguë, C.; Landry, H.; Peng, Q. Analysis of energy requirement for hemp fibre decortication using a hammer mill. Can. Biosystems Eng. 2012, 54, 2.1-2.8.

Placet, V.; Trivaudey, F.; Cisse, O.; Gucheret-Retel, V.; Lamine Boubakar, M. Diameter dependence of the apparent tensile modulus of hemp fibres: A morphological, structural or ultrastructural effect. Compos. Part A Appl. Sci. Manuf. 2012, 43, 275-287.

Baley, C. Influence of kink bands on the tensile strength of flax fibers. J. Mater. Sci. 2004, 39 (1), 331-334.

Grégoire, M.; Barthod-Malat, B.; Labonne, L.; Evon, Ph.; De Luycker, E.; Ouagne, P. Investigation of the potential of hemp fibre straws harvested using a combine machine for the production of technical load-bearing textiles. Ind. Crops Prod. 2020, 145, 111988.

If this mechanical extraction is used for the treatment of the plant materials from other regions, what are important issues that should be considered?

Answer:

The fibre extraction processes used in this study were selected to work with bast fibre type plants such as hemp, flax or nettle. As mentioned in the text, it is important that a minimum level of retting is performed so that the natural cements binding the fibres together and the ones binding the fibres to the rest of the plant are degraded. This favours the fibre division and the separation between the fibres and the woody core of the plant and the bark. As these machines are mainly dedicated to the fibre extraction of European bast fibres, one can imagine that they could also operate with other types of bast fibre plants such as kenaf or jute for example. Of course, modification of some of the settings would be necessary to accommodate the changes in plant diameter, or woody core breaking strength.

Both the machines were designed to operate in a dry state. If dried, resources such as banana trunks or sisal leaves could probably be used with the breaking rollers and the breaking card. However, some modifications in the corrugated rollers and of course in the process settings would probably be necessary. In any case, dedicated machines already exist for these kinds of resources (Snyder et al., 2006; Tenerife et al., 2019) and the machines presented in this study should be dedicated to the use of retted bast fibre plants.

This part was added in the text in the discussion Section (lines 742-757).

Related references:

Snyder, B.J.; Bussard, J.; Dolak, J.; Weiser, T. A portable sisal decorticator for Kenyan farmers. Int. J. Serv. Learning Eng. 2006, 2 (1), 92-116.

Tenerife, P.; De la Cruz, A.; Arce, A.; Pabularcon, Ma.; Ortega, K.; Rafallo, R. Design and development of banana fiber decorticator with wringer. Int. J. Recent Technol. Eng. 2019, 8 (1S4), 82-84.

Reviewer 2 Report

The present investigated the fiber extraction process with different dew retting durations and harvesting techniques to use them for producing innovative geotextiles. The chemical composition, fibre content and tensile properties of fibres were obtained. However, the paper subject was considered to be not within the scope of “Coatings”. The present journal devoted to the science and engineering of coatings, thin and thick films, surfaces and interfaces. Therefore, it was not appropriate to publish the present work.

Author Response

We thank Reviewer 2 for his helpful comments. Under are the answers to his questions.

The present investigated the fiber extraction process with different dew retting durations and harvesting techniques to use them for producing innovative geotextiles. The chemical composition, fibre content and tensile properties of fibres were obtained. However, the paper subject was considered to be not within the scope of “Coatings”. The present journal devoted to the science and engineering of coatings, thin and thick films, surfaces and interfaces. Therefore, it was not appropriate to publish the present work.

Answer:

We thank Reviewer #2 for his remark. This paper was submitted to the Coatings SI untitled "Natural Fiber Based Composites". It is in perfect accordance with the information mentioned on the link under concerning this specific SI.

https://www.mdpi.com/journal/coatings/special_issues/fiber_based

In particular, it is written that the papers to submit “will focus on cellulosic and lignocellulosic fibers, and on composite materials reinforced with renewable fibers”.

In the submitted paper, the results presented are indeed dealing with cellulosic fibres (i.e., natural fibres). And, the authors suggest that, once extracted, these natural fibres from oleaginous flax straw could be used for geotextile applications just as for composite ones.

Reviewer 3 Report

The research presented in the article is maybe not novel but for sure interesting and important in the context of sustainable development and for economic reasons.

The research aim is clearly defined and achieved. The scientific nature of the research was ensured by the selection of various experimental materials and the use of various extraction methods. The selection of the tested parameters and the methods of their determination is appropriate in the context of the goal.

Some doubts are raised by the presentation of the research results and discussion. In its present form, it is quite difficult to folow the results, due to the separate presentation of the data, and their discription in the next chapter. In fact, the "discussion" is too extensive, focused on the obvious observations presented in the tables for the individual parameters studied. While there is no in-depth analysis of the studied variables in a comprehensive approach. It is suggested to make a slight change in the structure of the article, in which the results and their discription will be combined, and the scientific discussion, including the reference to the results of other studies, will constitute a separate chapter.

The conclusions correspond well with the goal and summarize most of the results obtained. In the opinion of the reviewer, however, it could be possible to present a clear recommendation regarding the most effective extraction variant, type of fiber cutting, or length of the dew retting, etc.

I wish you good luck!

Author Response

We thank Reviewer 3 for his helpful comments. Under are the answers to his questions.

Some doubts are raised by the presentation of the research results and discussion. In its present form, it is quite difficult to follow the results, due to the separate presentation of the data, and their description in the next chapter. In fact, the "discussion" is too extensive, focused on the obvious observations presented in the tables for the individual parameters studied. While there is no in-depth analysis of the studied variables in a comprehensive approach. It is suggested to make a slight change in the structure of the article, in which the results and their description will be combined, and the scientific discussion, including the reference to the results of other studies, will constitute a separate chapter.

Answer:

On the one hand, the results and their description were combined in the same part (Results section) (lines 256-474). On the other hand, the Discussion section was thus logically reduced in length, and it was also completed in a scientific point of view with results from other studies and a scientific discussion (lines 475-757). Parts in the original version of the manuscript which were kept in the Discussion section were those considered by the authors as really discussing the results (especially with others from the literature) and not simply describing them. We hope that these changes will suit Reviewer #3.

The conclusions correspond well with the goal and summarize most of the results obtained. In the opinion of the reviewer, however, it could be possible to present a clear recommendation regarding the most effective extraction variant, type of fiber cutting, or length of the dew retting, etc.

Answer:

At the end of the Conclusion section, an additional paragraph was added with clear recommendations for future work regarding the dew retting duration, the type of fibre cutting (i.e., cutting height and disconnection of the combine harvester shredder), the most effective mechanical extraction equipment to use, and the need of the sieving extra-step to improve the purity in fibres of the final product (lines 783-792).